# Identification of Candidate Biomarkers of Alzheimer’s Disease via Multiplex Cerebrospinal Fluid and Serum Proteomics

**DOI:** 10.3390/ijms241814225

**Published:** 2023-09-18

**Authors:** Ping Liu, Lingxiao Li, Fangping He, Fanxia Meng, Xiaoyan Liu, Yujie Su, Xinhui Su, Benyan Luo, Guoping Peng

**Affiliations:** 1Department of Neurology, The First Affiliated Hospital, Zhejiang University School of Medicine, Hangzhou 310003, China; liuping908@163.com (P.L.); 13732315086@163.com (L.L.); hefangping@126.com (F.H.); mengfanxia@163.com (F.M.); liuxiaoyandongda@126.com (X.L.); suyuyu88@163.com (Y.S.); luobenyan@zju.edu.cn (B.L.); 2Department of Neuclear Medicine, The First Affiliated Hospital, Zhejiang University School of Medicine, Hangzhou 310003, China; suxinhui@zju.edu.cn

**Keywords:** Alzheimer’s disease, proteomes, differentially expressed proteins, bioinformatics analysis

## Abstract

Alzheimer’s disease (AD) is the most prevalent form of dementia among elderly people worldwide. Cerebrospinal fluid (CSF) is the optimal fluid source for AD biomarkers, while serum biomarkers are much more achievable. To search for novel diagnostic AD biomarkers, we performed a quantitative proteomic analysis of CSF and serum samples from AD and normal cognitive controls (NC). CSF and serum proteomes were analyzed via data-independent acquisition quantitative mass spectrometry. Our bioinformatic analysis was based on Gene Ontology (GO) functional annotation analysis and Kyoto Encyclopedia of Genes and Genomes (KEGG) enrichment. In comparison to the controls, 8 proteins were more abundant in AD CSF, and 60 were less abundant in AD CSF, whereas 55 proteins were more and 10 were less abundant in the serum samples. ATPase-associated activity for CSF and mitochondrial functions for CSF and serum were the most enriched GO terms of the DEPs. KEGG enrichment analysis showed that the most significant pathways for the differentially expressed proteins were the N-glycan biosynthesis pathways. The area under the curve (AUC) values for CSF sodium-/potassium-transporting ATPase subunit beta-1 (AT1B1), serglycin (SRGN), and thioredoxin-dependent peroxide reductase, mitochondrial (PRDX3) were 0.867 (*p* = 0.004), 0.833 (*p* = 0.008), and 0.783 (*p* = 0.025), respectively. A panel of the above three CSF proteins accurately differentiated AD (AUC = 0.933, *p* = 0.001) from NC. The AUC values for serum probable phospholipid-transporting ATPase IM (AT8B4) and SRGN were moderate. The AUC of the CSF SRGN + serum SRGN was 0.842 (*p* = 0.007). These novel AD biomarker candidates are mainly associated with inflammation, ATPase activity, oxidative stress, and mitochondrial dysfunction. Further studies are needed to investigate the molecular mechanisms by which these potential biomarkers are involved in AD.

## 1. Introduction

Alzheimer’s disease (AD), the world’s leading cause of dementia in the elderly, is characterized by clinically significant cognitive impairment and progressive neurodegeneration [1]. Amyloid-beta (Aβ) peptide accumulation is thought to be a significant early event in AD pathogenesis; however, AD is multifactorial and polygenic in nature, and Aβ may be necessary but insufficient to cause AD. The specific pathophysiology of this fatal condition is still unknown, and there are currently no proven effective treatments to reverse or hinder its progression. Other mechanisms have been hypothesized to be important factors in AD development and progression, including tau-mediated pathophysiology, neuroinflammation, mitochondrial dysfunction, axonal degeneration, synaptic dysfunction, and microvascular disease [2,3]. However, additional AD biomarkers are required to better understand the underlying mechanisms, track the progression and the degree of cognitive impairment, and identify targets for disease-modifying therapy.

Comprehensive proteomic analysis is complementary to genetic mapping and reveals another layer of molecular events in AD. It has enabled the profiling of brain tissue and biofluids at an unprecedented scale, resulting in the identification of potential biomarkers for subsequent validation and the construction of many novel hypotheses [4]. Cerebrospinal fluid (CSF) is a valuable source of potential proteomic biomarkers for both diagnosis and prognosis of AD. With the emergence and development of feasible blood-based arrays and high-throughput proteomic technologies, novel diagnostic and prognostic blood-based biomarkers for AD are now becoming a reality [5]. In AD subjects, some blood-based biomarkers change in synchrony with CSF biomarker values and may therefore be useful for the early and precise diagnosis, prognosis, and monitoring of disease progression and treatment effects [5,6].

A panel of dependable protein biomarkers rather than a single marker will aid the development of highly specific tests for early illness identification in high-risk populations. Such indicators might reveal unanticipated biological processes as well as brand-new prospective treatment targets for further research. Thus, to comprehensively search for novel diagnostic biomarkers, we performed a quantitative proteomic analysis of CSF and serum samples in AD patients and healthy subjects.

## 2. Results

### 2.1. Demographic and Clinical Characteristics of the Study Subjects

The demographic and clinical parameters of the AD and normal cognitive control (NC) groups are described in Table 1. There was no significant difference between the AD and NC groups in age, gender, years of education, diabetes, hypertension, or APOE genotype status. AD patients had significantly lower MMSE, MoCA, and Clock Drawing Test scores than the NCs. Meanwhile, the level of CSF Aβ42 was significantly lower, but the level of CSF t-tau was much higher in the AD group compared with the NC group.

### 2.2. Quantitative Proteomic Profiling of CSF and Serum Samples from AD and NC Subjects

A total of 22 CSF samples (12 AD and 10 NC) and 60 serum samples (30 AD and 30 NC) were analyzed using a DIA-MS approach. Among the revealed 8321 precursors, 7670 peptides, and 1308 proteins (Appendix A) in the CSF samples, the levels of 68 proteins were significantly different between the two groups. In the AD group, the levels of eight proteins—HV316 (gene name: *IGHV3-16*), PDCD5 (*PDCD5*), H2B1K (*H2BC12*), CAH11 (*CA11*), ACBP (*DBI*), CH60 (*HSPD1*), PPIC (*PPIC*), and MFAP5 (*MFAP5*)] were elevated, while 60 proteins [the top nine being: MYP2 (*PMP2*), AT1B1 (*ATP1B1*), TPM3 (*TPM3*), LIGO3 (*LIGO3*), SRGN (*SRGN*), MCP (*CD46*), PRDX3 (*PRDX3*)*,* CAZA2 (*CAPZA2*), and ZNF84 (*ZNF84*)—were downregulated compared with the NC group. Among the revealed 8564 precursors, 7597 peptides, and 968 proteins (Appendix A) in the serum samples, 65 DEPs were identified between the two groups. In the AD group, the levels of 55 proteins [the top nine being KVD13 (*IGKV1D-13*), FITM1 (*FITM1*), B3AT (*SLC4A1*), IGD, SRGN (*SRGN*), GTR1 (*SLC2A1*), STOM (*STOM*), K2C72 (*KRT72*), and EFCB6 (*EFCAB6*)] were upregulated, while the levels of 10 proteins were downregulated [LV151 (*IGLV1-51*), IGHM (*IGHM*), ALBU (*ALB*), OTUD4 (*OTUD4*), AT8B4 (*ATP8B4*), PHX2B (*PHOX2B*), FHR5 (*CFHR5*), CBLN4 (*CBLN4*), RBX2 (*RNF7*), and TEN1 (*TENM1*)] compared with the NC group. The volcano plots of these proteins are shown in Figure 1. The top nine upregulated and downregulated proteins are shown in Table 2 (CSF) and Table 3 (serum), as well as Figure 2 and Figure 3. It is worth noting that serglycin (SRGN) was markedly downregulated in AD CSF but upregulated in AD serum.

### 2.3. Bioinformatic Analysis of Specific Proteins Expressed in the AD Group

GO enrichment and KEGG pathway analyses were conducted to investigate the DEPs at the functional level. The top 10 GO biological processes, cellular components, and molecular functions (Figure 4A and Figure 5A) and the top 20 KEGG pathways (Figure 4B and Figure 5B) for CSF and serum DEPs were selected according to the number and significance of gene enrichment. The most enriched GO terms of the DEPs included ATPase-associated activity for CSF and mitochondrial functions for CSF, and serum were the most enriched GO terms of the DEPs. The N-Glycan biosynthesis pathways were the most significant KEGG pathways of the CSF DEPs.

### 2.4. Associations between DEP Levels and Cognitive Function

To examine the associations between DEP levels and cognitive function, we performed a linear correlation analysis. There were significant correlations between the MMSE, MoCA, and AVLT delayed recalling scores and most of those proteins.

As shown in Figure 6A, the level of CSF AT1B1 was positively associated with the AVLT immediate recall and short-term delayed recall (5 min delayed recall) scores. The level of CSF SRGN was positively associated with the AVLT short-term delayed recall score. The CSF PRDX3 level was positively associated with the AVLT short-term and long-term delayed recall (5-min, 20 min recall) and MMSE and MoCA scores (*p* all < 0.05).

For the serum DEPs, Figure 6B shows that AT8B4 was positively associated with the cognitive assessment scores, especially with the AVLT immediate, short-term, and long-term delayed recall scores (*p* all < 0.05).

### 2.5. Assessment of the Discriminative Ability of Candidate Biomarkers Using AUC Analysis

To evaluate the discriminative ability of the candidate DEPs, we performed ROC analysis. The AUC values for CSF AT1B1, SRGN, PRDX3 were 0.867 (95% confidence interval (CI): 0.684–1.000, *p* = 0.004), 0.833 (95% CI: 0.663–1.000, *p* = 0.008), and 0.783 (95% CI: 0.572–0.994, *p* = 0.025), respectively, indicating their potential value to distinguish AD from NCs. Notably, a combined panel of the above three CSF proteins accurately classified AD (AUC = 0.933, *p* = 0.001) from NCs (Figure 7A).

The AUC values for serum AT8B4 and SRGN were 0.758 (95% CI: 0.636–0.879, *p* = 0.001) and 0.602 (95% CI: 0.452–0.753, *p* = 0.008), respectively. The discriminative value for combined serum AT8B4 and SRGN was moderate (AUC = 0.797, 95% CI: 0.683–0.910, *p* < 0.001) (Figure 7B).

We also analyzed the discriminative efficiency of CSF SRGN combined with serum SRGN levels. The ROC curve showed that CSF SRGN plus serum SRGN was also an effective classifier between AD patients and NCs with an AUC of 0.842 (95% CI: 0.663–1.000, *p* = 0.007) (Figure 7C).

Finally, we compared the AUC data of AD diagnostic biomarkers and combined differentially expressed proteins in CSF. As shown in Figure 7D, the AUC value (95% CI: 0.883–1.000, *p* = 0.001) of CSF AT1B1 + SRGN + PRDX3 (0.933) was lower but close to those of CSF Aβ42, t-tau, and p-tau, which were 0.992 (95% CI: 0.965–1.000, *p* < 0.001), 0.958 (95% CI: 0.884–1.000, *p* = 0.008), and 1 (95% CI: 1.0–1.0, *p* < 0.001), respectively.

## 3. Discussion

Encouraging advances have been made in biofluid proteome profiling using MS-based approaches for the identification of AD biomarkers [4]. Numerous recent studies have characterized the proteomes of AD cerebrospinal fluid [6,7,8,9]. However, few deep MS-based profiling studies of AD plasma/serum samples have been reported [4]. To our knowledge, this is one of the first studies to perform deep MS-based proteomic profiling with respect to both CSF and serum samples from AD patients and NCs. The DEPs included but were not limited to PRDX3, SRGN, and AT1B1 in the CSF and SRGN and AT8B4 in the serum. These proteins are involved in inflammation, ATPase activity, oxidative stress, and mitochondrial dysfunction and may serve as novel AD biomarker candidates.

The reported MS-based proteomic studies, which primarily focused on CSF samples, discovered varying numbers of proteins associated with AD. Among studies that defined AD based on core CSF biomarkers (amyloid, t-tau, and p-tau), Sathe et al. presented a deep pilot CSF analysis from five AD subjects and five controls and identified 2327 proteins with 139 DEPs, including MAPT, NPTX2, VGF, GFAP, NCAM1, PKM, and YWHAG [7]. Higginbotham et al. identified 225 downregulated proteins and 303 upregulated proteins in 20 AD cases compared with 20 controls, including MAPT, NEFL, GAP43, FABP3, CHI3L1, NRGN, VGF, GDI1, and SMOC1 [9]. In three separate cohorts, Bader et al. examined CSF proteomics for a total of 109 control and 88 AD cases. The consistently upregulated or downregulated proteins in AD CSF included protein/nucleic acid deglycase DJ-1 (PARK7), superoxide dismutase 1 (SOD1), AD-like 14-3-3 protein ζ/δ (YWHAZ), YKL-40 [8]. Wang et al. recently performed an integrated analysis of ultra-deep proteomes of the cortex, CSF, and serum in AD [6]. They identified 37 proteins as potential AD biomarkers across these three tissues, among which 59% were mitochondrial proteins. They compared human CSF and 5 × FAD mouse CSF datasets, identifying six promising AD signature proteins: SMOC1, TAU, GFAP, SUCLG2, PRDX3, and NTN1 [6]. The diversity of AD-related CSF proteins indicates the complexity of AD pathophysiology and its multifaceted nature. On the other hand, numerous confounding factors, such as age, gender, ethnicity, sample size, specimen quality, different proteomics platforms and pre-analytical protocols, etc., may have an impact on the various DEPs discovered in the aforementioned studies [4,10].

In this study, except for the markedly decreased amyloid β-42 and increased t-tau and phosphorylated tau (p-tau) in CSF, we discovered significantly decreased levels of thioredoxin-dependent peroxide reductase, mitochondrial (PRDX3) in the CSF of AD subjects, which was consistent with Wang et al. [6]. PRDX3, also known as peroxiredoxin 3, belongs to a superfamily of peroxidases that function as protective antioxidant enzymes. Among the six peroxiredoxin isoforms (PRDX1–PRDX6), PRDX3 is the only one exclusively localized to mitochondria, which are the main source of reactive oxygen species. Excessive levels of reactive oxygen species are harmful to cells, inducing mitochondrial dysfunction, DNA damage, lipid and protein oxidation, and, ultimately, apoptosis. Neuronal damage induced by oxidative stress and mitochondrial dysfunction is associated with numerous neurodegenerative disorders, such as AD, Parkinson’s disease, and cerebellar ataxia [11]. Compared with APP transgenic mice, APP/PRDX3 transgenic mice show less cognitive decline and reduced amyloid beta levels in the brain [12]. PRDX3 knockdown in an AD cell model (N2a-APPswe cells) induced the dysregulation of more than one hundred proteins that were enriched for protein localization to the plasma membrane, the lipid catabolic process, and intermediate filament cytoskeleton organization [13]. Above all, the results indicate mitochondrial protein changes in AD brain. Further studies are required to understand the underlying mechanism.

It is important to note that a novel biomarker, SRGN, was markedly downregulated in the CSF and considerably elevated in the serum of AD patients in our investigation. SRGN, the only secretory vesicle proteoglycan, can carry heparin chains and is a core protein of heparan sulfate proteoglycans. The high negative charge of its heparan sulfate chains facilitates the packaging of positively charged proteases, histamine, and other inflammatory mediators [14]. SRGN plays an important regulatory role in the inflammatory response of the central nervous system. Heparan sulfate proteoglycans promote amyloid fibril formation and tau fibrillization in AD and provide resistance against proteolytic breakdown [15,16]. Recently, Lorente-Gea et al. [17] investigated the expression of heparan sulfate proteoglycans in various human AD brain areas at different Braak stages. SRGN and SDC4 were over-expressed in most of the areas, and immunohistochemistry revealed the presence of SRGN in all three types of AD lesion: neuritic plaques, cerebral amyloid angiopathy, and neurofibrillary pre-tangles and tangles. SRGN is the only intracellular heparan sulfate proteoglycan; therefore, the authors speculated that it may play a central role in AD stabilization and progression by means of the 3-O-sulfated domains in heparan sulfate polysaccharide chains [17]. Furthermore, significant numbers of the variously expressed proteins are involved in N-glycan biosynthesis, according to the functional interpretation provided by the KEGG analysis conducted for the present study. It is currently considered that N-glycans contribute significantly to the development of AD. Glycosylation, the most prevalent post-translational modification, is profoundly involved in a variety of diseases by altering the behaviors of important glycoproteins, including subcellular localization, enzyme activity, and protein–protein interactions. These findings, together with the fact that most AD-related molecules are modified with glycans, indicate that it is possible for glycosylation to regulate the genesis and progression of AD.

In the periphery, SRGN is mainly expressed in the storage granules and secretory vesicles of hematopoietic cells. It is involved in inflammation [18,19] and promotes angiogenesis in a variety of tumors [20]. In in vitro experiments, proinflammatory agents, lipopolysaccharide, and interleukin 1β (IL1β) increased the synthesis and secretion of SRGN, which is part of the inflammatory response activated by primary human endothelial cells and monocytes [19,21]. In normal mouse chondrocytes, lipopolysaccharide increased the expression of SRGN mRNA and protein and the production of the pro-inflammatory mediators, TNFα, IL1β, IL6, iNOS, and MMP9 through NF-κB activation. CD44 expression and the inflammatory response were significantly reduced by SRGN siRNA treatment. Wang et al. recently found that SRGN levels were dramatically increased in plasma samples from diabetic retinopathy cases compared with type 2 diabetes mellitus or healthy participants [22], and a very high percentage of patients with diabetic retinopathy develop AD [23].

Furthermore, sodium-/potassium-transporting ATPase subunit beta-1 (Na^+^-K^+^-ATP1B1) was downregulated in AD CSF in this study. Na^+^-K^+^-ATPase, also known as the Na^+^-K^+^-pump, transports three Na^+^ ions out of a cell and two K^+^ ions into a cell by hydrolyzing ATP. Na^+^-K^+^-ATPase is a hetero-oligomer composed of α subunits, β subunits, and a tissue-specific regulatory γ subunit. The α subunit plays an important catalytic role, whereas the β subunit mainly assists with the folding, targeting, and correct insertion of the newly synthesized α subunit into the cell membrane, thereby stabilizing the structure and regulating the activity of the α subunit [24,25]. In the central nervous system, Na^+^-K^+^-ATPase establishes transmembrane ion gradients and maintains the excitability of neurons and glial cells, thereby playing an important role in the process of learning and memory. The dysregulation and deficiency of neuronal Na^+^-K^+^-ATPase have been implicated in the pathogenesis of cognitive disorders, including vascular dementia and AD. The interaction of Aβ oligomers with Na^+^-K^+^-ATPase contributes significantly to the development of AD [26]. In another study, Na^+^-K^+^-ATPase expression was significantly decreased in patients with AD and in a transgenic mouse model of AD [27,28]. Moreover, our bioinformatics analysis showed that the most enriched GO term included ATPase-associated activity. Therefore, the decreased CSF level of β1 observed in this study indicated that neuronal Na^+^-K^+^-ATPase deficiency may be involved in the pathogenesis of AD.

Additionally, AT8B4, or probable phospholipid-transporting ATPase IM, which is encoded by *ATP8B4*, is a phospholipid transporter in the cell membrane and belongs to the cation transport ATPase (P-type) family and type IV subfamily (P4-ATPases). P4-ATPases utilize the energy from ATP hydrolysis to transport or flip aminophospholipids from the exocytoplasmic (extracellular/lumen) to the cytoplasmic leaflet of cellular membranes. Interestingly, the association between ATP8B4 and AD was indicated via a genome-wide association study [29]. Recently, a whole-exome sequencing study identified rare damaging variants in *ATP8B4* as a risk factor for AD [30]. *ATP8B4* is predominantly expressed in microglia in the brain; therefore, this result provides additional evidence for inflammation in AD.

In all, we found that a certain amount of DEPs, especially CSF AT1B1, SRGN and PRDX3, and serum AT8B4, were significantly correlated with the results of the cognitive assessment. The ROC analysis of CSF AT1B1, SRGN and PRDX3, and serum AT8B4 and SRGN demonstrated good discriminating efficacy. Better predictive value is provided by combining selected proteins. Combined AT1B1, SRGN, and PRDX3 in CSF was better at differentiating AD subjects from the controls compared with any of the individual markers.

There are two main limitations of this study that should be noted. Firstly, not all AD patients underwent lumbar puncture; therefore, the number of CSF samples was relatively small. We did not find a statistically significant difference in serum tau or APP-related proteins between the AD and control groups, and this can be attributed to the limited sample size and diversity amongst patients. Secondly, the biological mechanisms of these potential proteomic markers have not been verified in follow-up cohorts or by molecular analysis.

## 4. Materials and Methods

### 4.1. Study Subjects

A total of 60 subjects [AD = 30, healthy normal cognition controls (NC) = 30] were included. The AD patients were recruited consecutively from the Memory Clinic, Department of Neurology at the First affiliated hospital, Zhejiang University School of Medicine.

They were diagnosed according to the criteria of the Diagnostic and Statistical Manual (DSM)-IV and of the International Working Group (IWG) for New Research Criteria for the diagnosis of AD, revised in 2014 (IWG-2) [31]. NCs were recruited in the same proportions of gender and age as the AD group.

Each participant underwent complete medical history evaluation and neurological and neuropsychological assessments, including Mini Mental State Examination (MMSE) [32], the Beijing version of the Montreal Cognitive Assessment (MoCA) [33], the Clock Drawing Test, the Auditory Verbal Learning Test–Huashan version (AVLT) [34], Clinical Dementia Rating, brain imaging, and standard laboratory tests. Apolipoprotein E (APOE) genotyping was performed to determine the presence of the ε2, ε3, and ε4 alleles following single-nucleotide polymorphisms: 3937T > C (rs429358) and 4075C > T (rs7412) [35].

Overall, 12 AD patients and 10 controls underwent lumbar puncture for CSF collection, and the other 18 underwent Florbetapir F-18 (18F-AV-45) positron emission tomography (PET)-computed tomography or PET–magnetic resonance scanning. A low CSF Aβ42 level or a positive 18F-AV-45 PET scan, which showed a high binding affinity and specificity to Aβ plaques, were additional requirements for inclusion in the AD group to confirm that patients had amyloid deposition in the brain [36].

The following were the participation exclusion criteria: other causes of cognitive impairment; severe malnutrition, infection, drug, or alcohol addictions within the past year; schizophrenia, schizoaffective disorder, or primary affective disorder; severe heart, brain, liver, kidney, lung, and hematopoietic system diseases; severe auditory, visual, or motor deficits impairing cognitive testing; and other serious primary diseases.

All participants or their legally authorized caregivers were informed of the purpose of this study and provided written informed consent. The study was approved by the ethics committee of the First affiliated hospital, Zhejiang University School of Medicine, Hangzhou, China (2016-035).

### 4.2. Clinical Samples

Peripheral blood was obtained from all participants. Serum samples were separated via centrifugation at 1000× *g* for 10 min at 4 °C and then stored in aliquots at −80 °C. Twelve AD patients and ten NCs underwent lumbar puncture, and the CSF samples were also collected in polypropylene vials, separated via centrifugation, and stored at −80 °C.

### 4.3. Mass Spectrometry (MS) Analysis

SDT buffer (4% SDS, 100 mM Tris-HCl, pH 7.6) was added to the CSF and serum samples, and the samples were boiled and centrifuged. The protein levels in the supernatant were quantified using a BCA Protein Assay Kit (P0012, Beyotime, Haimen, China). The peptides were diluted to a final concentration of 10 ng/μL in 0.1% formic acid (FA). For each sample, 200 ng of protein was loaded on an Evotip for desalting and then washed with 20 μL 0.1% FA followed by the addition of 150 μL storage solvent (0.1% FA) to keep the Evotip wet until analysis. The samples were then analyzed using an Evosep One system (Evosep, Odense, Denmark) coupled to a timsTOF Pro (Bruker, Bremen, Germany) equipped with a CaptiveSpray source. The timsTOF Pro (Bruker, Bremen, Germany) was operated in parallel accumulation–serial fragmentation (PASEF) mode. For diaPASEF, we adapted the instrument firmware to perform the data-independent isolation of multiple precursor windows within a single trapped ion mobility spectrometry (TIMS) separation (100 ms). We used a method with two windows in each 100 ms diaPASEF scan. Eighteen of these scans covered the diagonal scan line for doubly and triply charged peptides in the *m*/*z*–ion mobility plane with narrow 25 *m*/*z* precursor windows.

Raw data-dependent acquisition (DDA) data were processed and analyzed via the use of Spectronaut (Biognosys AG, Schlieren, Switzerland) using the default settings to generate an initial target list. Spectronaut was configured to search the Uniprot_HomoSapiens_20394_20210127 database, assuming trypsin as the digestion enzyme. Carbamidomethylation was specified as the fixed modification. Oxidation and acetylation (Protein N-term) were specified as the variable modifications. A Q value cutoff of 1% was applied to precursor and protein levels. Raw data-independent acquisition (DIA) data were processed and analyzed via the use of Spectronaut (Biognosys AG, Schlieren, Switzerland) using the default settings. A Q value cutoff of 1% was applied to precursor and protein levels. Decoy generation was set to ‘mutated’, which is similar to ‘scrambled’ but only applies a random number of AA position swamps (min = 2, max = length/2). The average top three filtered peptides that passed the 1% Q value cutoff were used to calculate the major group quantities. After applying Student’s *t* test, differentially expressed proteins (DEPs) were selected if their *p* value was <0.05 and fold change was ≥1.2.

### 4.4. Bioinformatic Analysis

All protein sequences were aligned to the *Homo sapiens* database, which was downloaded from NCBI (ncbi-blast-2.2.28+-win32.exe). Only the top 10 sequences with E-values ≤ 1 × 10^−3^ were retained. Secondly, the GO term (database version: go_201504.obo) of the sequence with the top Bit-Score was selected by Blast2GO. Then, the annotation of the proteins using GO terms was completed by using Blast2GO Command Line. After initial elementary annotation, InterProScan was used to search the European Bioinformatics Institute (EBI) database by motif to add the functional information of the motifs to the proteins to improve our annotation. Further improvements in annotation and connection between the GO terms were then carried out using ANNEX. Fisher’s exact test was used to enrich the GO terms by comparing the number of differentially expressed proteins and total proteins correlated with the GO terms. Pathway analysis was performed using the KEGG database. Fisher’s exact test was used to identify the significantly enriched pathways by comparing the number of DEPs and total proteins correlated to pathways.

### 4.5. Statistical Analysis

Statistical analyses were performed using SPSS version 23.0 (IBM Corp., Armonk, NY, USA). The normality of distribution of continuous variables such as age, MMSE score, and protein concentration was tested via a one-sample rendition of the Kolmogorov–Smirnov test. The means of the continuous normally distributed variables of the two groups were compared by using Student’s *t* test on independent samples. The Mann–Whitney U test was used to compare the means of variables not normally distributed. Group differences in categorical data, such as sex and apolipoprotein genotype subgroup, were analyzed using the Chi-square test. Correlative analysis was performed using a linear regression model. To assess the potential prediction efficacy for selected DEPs and in combinations, receiver operating characteristic (ROC) curve and area under the ROC curve (AUC) analysis were performed. *p* < 0.05 was considered as statistically significant.

## 5. Conclusions

We identified DEPs in the CSF and serum of AD subjects compared with healthy controls, which were mainly associated with inflammation, ATPase activity, oxidative stress, and mitochondrial dysfunction. These potential biomarkers could be of great importance to classify AD from non-AD individuals, especially when combined together. The potential proteomic markers identified in our study may therefore be important for developing early biomarkers for AD and for understanding its multidimensional pathogenesis. Further studies are needed to explore the biological mechanisms of these potential biomarkers in AD.

## Figures and Tables

**Figure 1 ijms-24-14225-f001:**
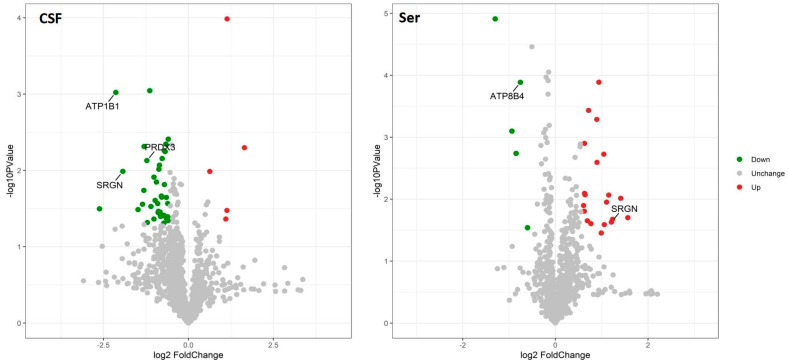
Volcano plots showing the distribution of differentially expressed proteins. The *x*-axis indicates fold change, and the *y*-axis indicates minus the log of *p*-value. Significantly upregulated and downregulated differentially expressed proteins are shown in red and green, respectively. Log2 fold change and significance cutoffs for differential expression were |1.2| and 0.05, respectively.

**Figure 2 ijms-24-14225-f002:**
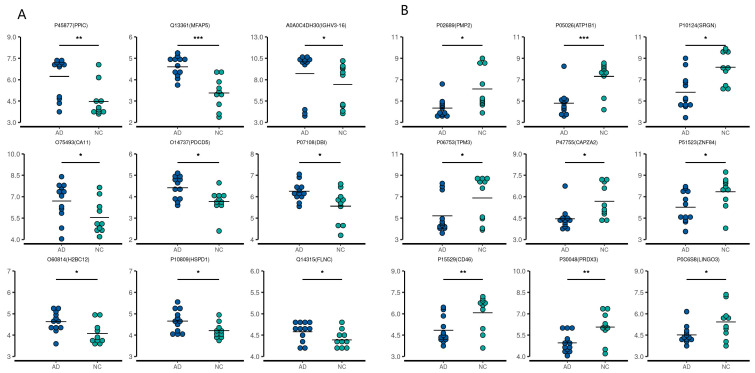
First nine upregulated (**A**) and downregulated (**B**) differentially expressed proteins in the CSF of AD patients compared with the controls. * *p* < 0.05, ** *p* < 0.01, *** *p* < 0.001.

**Figure 3 ijms-24-14225-f003:**
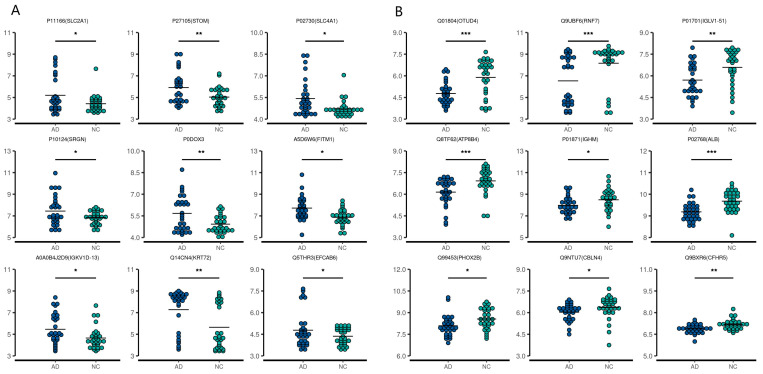
First nine upregulated (**A**) and downregulated (**B**) differentially expressed proteins in the serum from AD patients compared with the controls. * *p* < 0.05, ** *p* < 0.01, *** *p* < 0.001.

**Figure 4 ijms-24-14225-f004:**
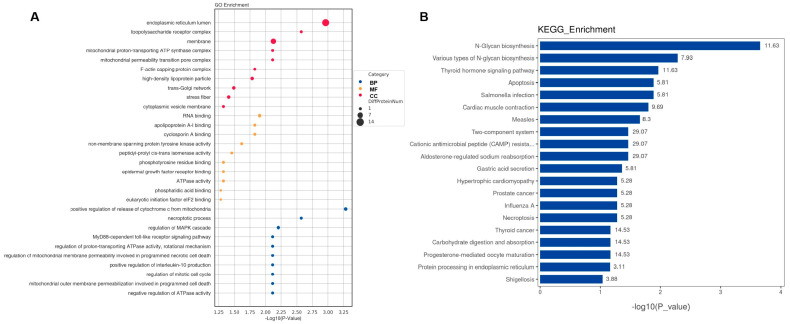
Functional analyses of differentially expressed proteins in CSF. (**A**) Bubble chart of GO enrichment analysis. (**B**) KEGG signal pathway analysis. GO, Gene Ontology; KEGG, Kyoto Encyclopedia of Genes and Genomes; BP, biological process; MF, molecular function; CC, cellular component; bubble size, the number of targets enriched pathway terms.

**Figure 5 ijms-24-14225-f005:**
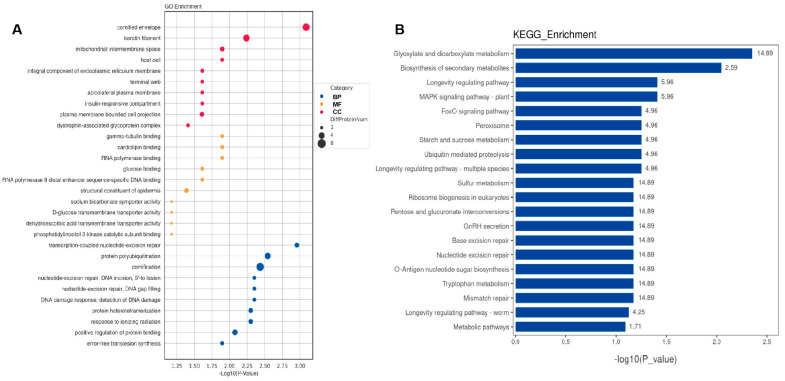
Functional analyses of differentially expressed proteins in serum. (**A**) Bubble chart of GO enrichment analysis. (**B**) KEGG signal pathway analysis. GO, Gene Ontology; KEGG, Kyoto Encyclopedia of Genes and Genomes; BP, biological process; MF, molecular function; CC, cellular component; bubble size, the number of targets enriched pathway terms.

**Figure 6 ijms-24-14225-f006:**
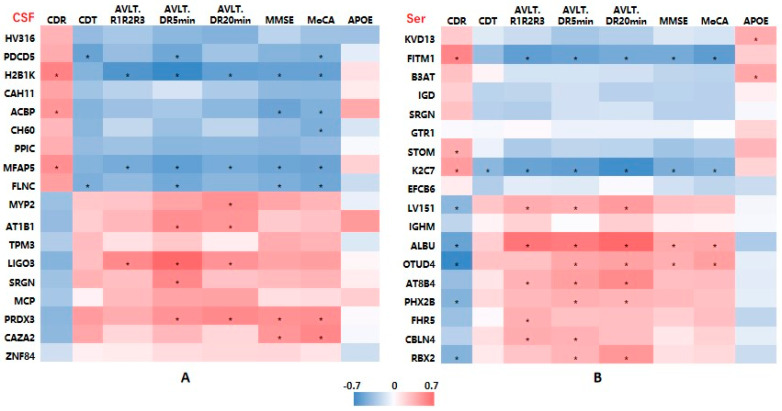
Heatmaps showing correlations between differentially expressed protein levels and clinical characteristics of AD patients. (**A**) CSF samples. (**B**) Serum samples. Abbreviations: AD, Alzheimer’s disease; MMSE, Mini-Mental State Examination; MoCA, Montreal Cognitive Assessment; CDR, Clinical Dementia Rating; CDT, clock drawing test; AVLT, Auditory-Verbal Learning Test, AVLT R1R2R3: mean score of three times of AVLT immediate recall; DR, delayed recall. Red means positive correlation, and blue means negative correlation; * *p* < 0.05.

**Figure 7 ijms-24-14225-f007:**
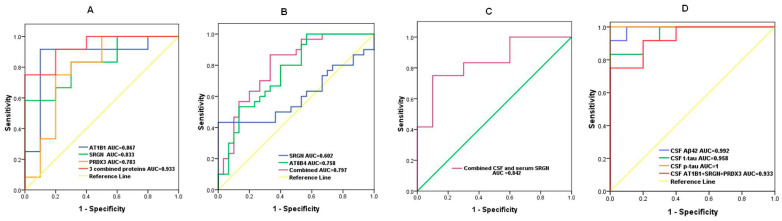
ROC curves showing the discriminative efficiency of selected DEPs to classify AD from controls. (**A**) AUC for CSF AT1B1, SRGN, PRDX3 levels and AT1B1 + SRGN + PRDX3; (**B**) AUC for serum AT8B4, SRGN, and AT8B4 + SRGN; (**C**) AUC for CSF SRGN+ serum SRGN. (**D**) AUC for CSF Aβ42, t-tau, p-tau, and AT1B1 + SRGN + PRDX3; ROC, receiver operating characteristic; DEPs, differentially expressed proteins; AUC, area under the curve.

**Table 1 ijms-24-14225-t001:** Demographic and clinical parameters of the two groups.

	AD (*n* = 30)	NC (*n* = 30)	t/χ^2^	*p* Value
Age (years)	65.17 ± 11.34	65.63 ± 7.08	0.246	0.620
Female (%)	14 (46.67%)	13 (43.33%)	0.067	0.795
Education (years)	9.07 ± 4.08	8.47 ± 3.15	0.346	0.557
Diabetes (%)	9 (30.0%)	8 (26.67%)	8.208	0.774
Hypertension (%)	11 (36.67%)	10 (33.33%)	7.326	0.786
MMSE	18.17 ± 6.10	26.43 ± 1.63	35.101	<0.001
MoCA	13.93 ± 5.27	23.70 ± 2.73	38.823	<0.001
CDT	19.96 ± 4.56	24.78 ± 2.61	116.43	<0.001
CDR	1.15 ± 0.51	0.00 ± 0.00	-	-
CSF Aβ42 (pg/mL)	452.99 ± 101.25	1062.29 ± 101.25	−5.495	<0.001
CSF Aβ40 (pg/mL)	9902.66 ± 2431.47	12,280.38 ± 1904.07	−2.513	0.021
CSF Aβ42/Aβ40	0.047 ± 0.011	0.086 ± 0.019	−6.029	<0.001
CSF t-tau (pg/mL)	413.27 ± 166.92	81.27 ± 82.79	5.714	<0.001
CSF p-tau (pg/mL)	195.1 ± 233.98	42.86 ± 16.13	2.045	0.054
APOE genotype group			3.314	0.191
E2	7 (23.3%)	13 (43.3%)		
E3	12 (40.0%)	11 (36.7%)		
E4	11 (36.7%)	6 (20.0%)		

Notes: Data are given as means ± SD or percentages. Twelve AD patients and ten controls underwent lumbar puncture, and the typical CSF Aβ42, Aβ40, t-tau, and p-tau were tested. There is no e2e4 genotype in this study. Abbreviations: AD, Alzheimer’s disease; NC, normal cognition healthy controls. MMSE, Mini-Mental State Examination; MoCA, Montreal Cognitive Assessment; CDT, clock drawing test; CDR, Clinical Dementia Rating; CSF, Cerebrospinal fluid; APOE, Apolipoprotein E. SD, standard deviation. APOE genotype group defined as E2 = e2e2 or e2e3; E3 = e3e3; and E4 = e3e4 or e4e4.

**Table 2 ijms-24-14225-t002:** Identification of differentially expressed proteins in the CSF from AD patients compared with healthy controls.

Accession Number	Protein Name	Protein	Gene Name	Change	*p* Value
A0A0C4DH30	Probable non-functional immunoglobulin heavy variable 3–16	HV316	*IGHV3-16*	↑	0.000546667
O14737	Programmed cell death protein 5	PDCD5	*PDCD5*	↑	0.005184637
O60814	Histone H2B type 1-K	H2B1K	*H2BC12*	↑	0.007706908
O75493	Carbonic anhydrase-related protein 11	CAH11	*CA11*	↑	8.88922 × 10^−26^
P07108	Acyl-CoA-binding protein	ACBP	*DBI*	↑	3.75501 × 10^−59^
P10809	60 kDa heat shock protein, mitochondrial	CH60	*HSPD1*	↑	1.4658 × 10^−25^
P45877	Peptidyl-prolyl cis-trans isomerase C	PPIC	*PPIC*	↑	0.003508082
Q13361	Microfibrillar-associated protein 5	MFAP5	*MFAP5*	↑	5.04183 × 10^−8^
Q14315	Filamin-C	FLNC	*FLNC*	↑	0.000943945
P02689	Myelin P2 protein	MYP2	*PMP2*	↓	5.57 × 10^−24^
P05026	Sodium/potassium-transporting ATPase subunit beta-1	AT1B1	*ATP1B1*	↓	0.000577
P06753	Tropomyosin alpha-3 chain;	TPM3	*TPM3*	↓	1.02 × 10^−16^
P0C6S8	Leucine-rich repeat and immunoglobulin-like domain-containing nogo receptor-interacting protein 3	LIGO3	*LINGO3*	↓	0.002143
P10124	Serglycin	SRGN	*SRGN*	↓	1.11 × 10^−48^
P15529	Membrane cofactor protein	MCP	*CD46*	↓	1.5 × 10^−26^
P30048	Thioredoxin-dependent peroxide reductase, mitochondrial	PRDX3	*PRDX3*	↓	0.000903
P47755	F-actin-capping protein subunit alpha-2	CAZA2	*CAPZA2*	↓	7.82 × 10^−11^
P51523	Zinc finger protein 84	ZNF84	*ZNF84*	↓	4.79 × 10^−20^

Note: This table shows the first nine upregulated and downregulated differentially expressed proteins.

**Table 3 ijms-24-14225-t003:** Identification of differentially expressed proteins in the serum from AD patients compared with healthy controls.

Accession Number	Protein Name	Protein	Gene Name	Change	*p* Value
A0A0B4J2D9	Immunoglobulin kappa variable 1D-13	KVD13	*IGKV1D-13*	↑	4.98 × 10^−21^
A5D6W6	Fat storage-inducing transmembrane protein 1	FITM1	*FITM1*	↑	1.91 × 10^−89^
P02730	Band 3 anion transport protein	B3AT	*SLC4A1*	↑	3.33 × 10^−75^
P0DOX3	Immunoglobulin delta heavy chain	IGD		↑	3.06 × 10^−55^
P10124	Serglycin	SRGN	*SRGN*	↑	1.95 × 10^−49^
P11166	Solute carrier family 2, facilitated glucose transporter member 1	GTR1	*SLC2A1*	↑	0.00029
P27105	Stomatin	STOM	*STOM*	↑	4.1 × 10^−101^
Q14CN4	Keratin, type II cytoskeletal 72	K2C72	*KRT72*	↑	0.00112
Q5THR3	EF-hand calcium-binding domain-containing protein 6	EFCB6	*EFCAB6*	↑	0.000927
P01701	Immunoglobulin lambda variable 1–51	LV151	*IGLV1-51*	↓	2.12 × 10^−58^
P01871	Immunoglobulin heavy constant mu	IGHM	*IGHM*	↓	1.8 × 10^−107^
P02768	Albumin	ALBU	*ALB*	↓	1.1 × 10^−124^
Q01804	OTU domain-containing protein 4	OTUD4	*OTUD4*	↓	8.57 × 10^−5^
Q8TF62	Probable phospholipid-transporting ATPase IM	AT8B4	*ATP8B4*	↓	3.41 × 10^−25^
Q99453	Paired mesoderm homeobox protein 2B	PHX2B	*PHOX2B*	↓	1.17 × 10^−57^
Q9BXR6	Complement factor H-related protein 5	FHR5	*CFHR5*	↓	1.5 × 10^−105^
Q9NTU7	Cerebellin-4	CBLN4	*CBLN4*	↓	3.24 × 10^−26^
Q9UBF6	RING-box protein 2	RBX2	*RNF7*	↓	0.000191

Note: This table shows the first nine upregulated and downregulated differentially expressed proteins.

## Data Availability

Data are available from the corresponding author upon reasonable request owing to privacy and ethical restrictions.

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
