# Peer review of "Identification of Candidate Biomarkers of Alzheimer’s Disease via Multiplex Cerebrospinal Fluid and Serum Proteomics"

_ijms, 2023, doi:10.3390/ijms241814225_

Round 1
Reviewer 1 Report
The work “Identification of candidate biomarkers of Alzheimer's disease by multiplex cerebrospinal fluid and serum proteomics” addressed both the CSF and serum AD proteome. This is relevant however I have some suggestions to improve the manuscript.
1) Lines 30 to 32 (abstract): I suggest to simply the sentence.
2) Lines 32 and 33 (abstract): “The most enriched GO terms included ATPase associated activity and mitochondrial functions”. Please clarify in the abstract if this is applicable for both serum and CSF proteome.
3) Line 142 (methods): What was the statistical test used to evaluate data distribution for further selection of t-test or Mann-Whitney?
4) Results – Table 2 and Table 3 – Please add columns with average abundance values of each protein in Controls and AD cases.
5) Results – In lines 165 to 167, authors indicate 8 proteins increased in AD group and enumerate their gene names however table 2 presented these 8 proteins plus Filamin-C. Please explain this inconsistency.
6) Results - In section 3.4, authors highlighted some differentially expressed proteins associated with cognitive assessment scores. However, other targets such as H2B1K and MFAP5 in CSF correlated with a higher number of cognitive tests. The same happened for serum and the proteins FITM1 and K2C7. Why had the authors not highlighted and validated these targets?
7) Results - Please include in figure 4 heatmaps all the differentially expressed proteins and not only some.
8) Results - Lines 233 and 234: “…combined value for AT8B4 and SRGN was moderate (AUC=0.793)…”. I believe it should be corrected for AUC=0.797
9) Results – Lines 245: “… AUC value of CSF AT1B1+SRGN+PRDX3…”. Please add the AUC value.
10) Results – Line 247 – AUC value of 1 refers to which protein? I believe it is P-Tau, clarify in the manuscript. Also include the number of individuals in which these proteins were assessed and which values were used to construct the ROC curves.
11) Results – I suggest validating the results for SRGN, AT1B1, PRDX3 and AT8B4 by ELISA to be able to conclude more accurately about their value as biomarkers.
12) Discussion – Lines 343 and 344 – “… we found that the majority of DEPs, especially CSF AT1B1, SRGN and PRDX3 and serum AT8B4 were significantly correlated with the results of the cognitive assessment…”. In my opinion, this sentence should be improved since, as I have already commented above, other targets correlated with more cognitive score scales. Also, heatmap 4 is not reflecting all DEPs but only some and, thus, may be the “majority” term does not apply.
13) I suggest to provide as supplementary material, MS files presenting abundances for all proteins.
Author Response
Dear Professor,
On behalf of my co-authors, we thank you very much for giving us an opportunity to revise our manuscript. We have carefully evaluated the critical comments and thoughtful suggestions, responded to these suggestions point-by-point, and revised the manuscript accordingly. Regarding your comments and suggestions, we wish to reply as follows:
The work “Identification of candidate biomarkers of Alzheimer's disease by multiplex cerebrospinal fluid and serum proteomics” addressed both the CSF and serum AD proteome. This is relevant however I have some suggestions to improve the manuscript.
1)Lines 30 to 32 (abstract): I suggest to simply the sentence.
Answer: Thanks for the reviewer’s suggestion. We simplified the sentence as “In comparison to controls, eight proteins were more abundant and 60 were less abundant in AD CSF, whereas 55 proteins were more and 10 were less abundant in serum samples.”
2) Lines 32 and 33 (abstract): “The most enriched GO terms included ATPase associated activity and mitochondrial functions”. Please clarify in the abstract if this is applicable for both serum and CSF proteome.
Answer: Thanks for the carefulness of the reviewer. We checked our results and rewrote the sentence as “ATPase-associated activity for CSF and mitochondrial functions for CSF and serum were the most enriched GO terms of the DEPs”.
3) Line 142 (methods): What was the statistical test used to evaluate data distribution for further selection of t-test or Mann-Whitney?
Answer: Thanks for the reviewer’s suggestion. We fixed this in our revised manuscript.
“Statistical analyses were performed using SPSS version 23.0 (IBM Corp., Armonk, NY). The normality of distribution of continuous variables such as age, MMSE score and protein concentration, was tested by one-sample Kolmogorov-Smirnov test. Means of continuous normally distributed variables of the two groups were compared by independent samples Student’s t test. Mann-Whitney U test were used to compare means of variables not normally distributed”.
4) Results – Table 2 and Table 3 – Please add columns with average abundance values of each protein in Controls and AD cases.
Answer: Thanks for the reviewer’s suggestion. We add the scatter plot in our revised manuscript. See figure 2 (A,B) and figure 3 (A,B).
Figure 2. First 9 upregulated (A) and downregulated (B) differentially expressed proteins in the CSF of AD compared with controls.
Figure 3. First 9 upregulated (A) and downregulated (B) differentially expressed proteins in the serum of AD compared with controls.
5) Results – In lines 165 to 167, authors indicate 8 proteins increased in AD group and enumerate their gene names however table 2 presented these 8 proteins plus Filamin-C. Please explain this inconsistency.
Answer: Thanks for the carefulness of the reviewer. Differentially expressed proteins (DEPs) were selected if their P value was < 0.05 and fold change was ≥1.2. We checked the statistical results, and found that the first 8 protein in the table 2 meets the two major standards mentioned above. The P value for Filamin-C was 0.0487(< 0.05),while its fold change was 1.14(<1.2). Table 2 shows the first 9 upregulated and downregulated differentially expressed proteins, so the Filamin-C was included.
6) Results - In section 3.4, authors highlighted some differentially expressed proteins associated with cognitive assessment scores. However, other targets such as H2B1K and MFAP5 in CSF correlated with a higher number of cognitive tests. The same happened for serum and the proteins FITM1 and K2C7. Why had the authors not highlighted and validated these targets?
Answer: In this article, we analyzed and discussed the proteins that exhibit elevated and/or decreased expression in both serum and cerebrospinal fluid, taking into account the multiple pathogenesis of AD. Meanwhile, we have conducted a series of literature searches on these you mentioned differently expressed proteins and discovered no reports on these proteins in Alzheimer's disease. But as you suggested, more samples will be needed in future studies for testing, analysis, and data validation.
7) Results - Please include in figure 4 heatmaps all the differentially expressed proteins and not only some.
Answer: Thanks for the reviewer’s suggestion. We once attempted to create a heat map of all differentially expressed proteins, but due to the large number of differentially expressed proteins and the low content of unlisted proteins, the results of the heat map were difficult to understand. Therefore, as shown in the heatmap figure, we demonstrate the correlation between the proteins with the most significant differential expression and the cognitive function.
8) Results-Lines 233 and 234: “combined value for AT8B4 and SRGN was moderate (AUC=0.793)”. I believe it should be corrected for AUC=0.797
Answer: We fixed this in our revised manuscript. We are so sorry to make this mistake.
9) Results – Lines 245: “… AUC value of CSF AT1B1+SRGN+PRDX3”. Please add the AUC value.0.933
Answer: Thanks for the reviewer’s suggestion. We added this in our revised manuscript as “As show in figure 5D, the AUC value 0.933 (95% CI: 0.883–1.000, P= 0.001) of CSF AT1B1+SRGN+PRDX3 was lower but close to those of CSF Aβ42”.
10)Results – Line 247 – AUC value of 1 refers to which protein? I believe it is P-Tau, clarify in the manuscript. Also include the number of individuals in which these proteins were assessed and which values were used to construct the ROC curves.
Answer: Thank you, and the AUC value of 1 refers to p-tau. We have stated “As show in figure 5D, the AUC value 0.933 (95% CI: 0.883–1.000, P = 0.001) of CSF AT1B1+SRGN+PRDX3 was lower but close to those of CSF Aβ42, t-tau and p-tau, which were 0.992 (95% confidence interval (CI): 0.965–1.000, P < 0.001), 0.958 (95% CI: 0.884–1.000, P = 0.008) and 1 (95% CI: 1.0–1.0, P < 0.001), respectively. ” in our manuscript.
Actually, in the methods and materials section, we expounded that 12 AD and 10 controls underwent lumbar puncture, and the typical CSF AD biomarkers, including Aβ42, Aβ40, t-tau and p-tau were tested.
11) Results – I suggest validating the results for SRGN, AT1B1, PRDX3 and AT8B4 by ELISA to be able to conclude more accurately about their value as biomarkers.
Answer: Thanks for the reviewer’s suggestion. We are collecting more samples for further verification. And as a limitation, we have stated that in the final discussion part of our manuscript, “Secondly, the biological mechanisms of these potential proteomic markers have not been verified in follow up cohorts or by molecular analysis”.
12) Discussion – Lines 343 and 344 – “… we found that the majority of DEPs, especially CSF AT1B1, SRGN and PRDX3 and serum AT8B4 were significantly correlated with the results of the cognitive assessment…”. In my opinion, this sentence should be improved since, as I have already commented above, other targets correlated with more cognitive score scales. Also, heatmap 4 is not reflecting all DEPs but only some and, thus, may be the “majority” term does not apply.
Answer: Thanks for the reviewer’s suggestion. We have re-written this part according to the Reviewer’s suggestion as follows, “we found that a certain amount of DEPs, especially CSF AT1B1, SRGN and PRDX3 and serum AT8B4, were significantly correlated with the results of the cognitive assessment.”
13) I suggest to provide as supplementary material, MS files presenting abundances for all proteins.
Answer: According to the reviewer’s suggestion, we provide the abundances for all proteins as supplementary materials.
We would like to express our great appreciation to you for comments on our paper. Looking forward to hearing from you.
Thank you and best regards.
Yours sincerely,
Guoping Peng, MD, PhD

Reviewer 2 Report
Research on AD biomarkers is very important, but discriminative descriptions are essential because there are already so many studies on AD biomarkers.
1. It is necessary to clarify the method section more clearly. Information on the number of CSF (12/10) and blood sampling in AD and NC is scattered throughout the text, and it is recommended to clearly organize them in study subjects. In addition, it is necessary to display the CSF values presented clearly in Table 1 so that it can be confirmed that the values are for 12 AD and 10 NC.
2. It seems necessary to describe the presence and types of comorbidities in 30 AD and 30 NC subjects. It was described that most of the biomarkers discovered were related to inflammation, but it is necessary to mention comorbidities to confirm if they are only related to AD pathology.
3. There is no problem with statistical analysis, but mention whether the version of SPSS is too low, and more than that, describe the source of the statistical program.
4. This study is referred to as "The majority of DEPs in the CSF and serum of AD subjects which were mainly associated with inflammation, ATPase activity, oxidative stress and mitochondrial dysfunction, were significantly correlated with the results of the cognitive assessment.". But it seems that more discussion is needed on the relationship between cognitive decline and the corresponding mechanism in the biomarker found.
Author Response
Dear Professor,
On behalf of my co-authors, we thank you very much for giving us an opportunity to revise our manuscript. We have carefully evaluated the critical comments and thoughtful suggestions, responded to these suggestions point-by-point, and revised the manuscript accordingly. Regarding your comments and suggestions, we wish to reply as follows:
Research on AD biomarkers is very important, but discriminative descriptions are essential because there are already so many studies on AD biomarkers.
1)It is necessary to clarify the method section more clearly. Information on the number of CSF (12/10) and blood sampling in AD and NC is scattered throughout the text, and it is recommended to clearly organize them in study subjects. In addition, it is necessary to display the CSF values presented clearly in Table 1 so that it can be confirmed that the values are for 12 AD and 10 NC.
Answer: Thanks for the reviewer’s suggestion. We revised the“Materials and Methods ”section,and we rewrote as follows:
“Sixty subjects [AD = 30, healthy normal cognition controls (NC) = 30] were included. The AD patients were recruited consecutively from the Memory Clinic, Department of Neurology at the First affiliated hospital, Zhejiang University School of Medicine. They were diagnosed according to the criteria of the Diagnostic and Statistical Manual (DSM)-IV and of the International Working Group (IWG) for New Research Criteria for the diagnosis of AD, revised in 2014 (IWG-2)[31]. NCs were recruited in the same proportions of gender and age as the AD group.
Each participant underwent complete medical history evaluation, neurological and neuropsychological assessments, including Mini Mental State Examination (MMSE)[32], the Beijing version of the Montreal Cognitive Assessment (MoCA)[33], the Clock Drawing Test, the Auditory Verbal Learning Test–Huashan version (AVLT)[34], Clinical Dementia Rating, brain imaging, and standard laboratory tests. Apolipoprotein E (APOE) genotyping was performed to determine the presence of the ε2, ε3, and ε4 alleles following single nucleotide polymorphisms:3937T>C (rs429358) and 4075C>T (rs7412) [35].
Twelve AD patients and 10 controls underwent lumbar puncture for CSF collection, and the other 18 underwent Florbetapir F-18 (18F-AV-45) positron emission tomography (PET)-computed tomography or PET-magnetic resonance scanning. A low CSF Aβ42 level or a positive 18F-AV-45 PET scan, which showed a high binding affinity and specificity to Aβ plaques, were additional requirements for inclusion in the AD group to confirm that patients had amyloid deposition in the brain[36].
Following were the participation exclusion criteria: other causes of cognitive impairment; severe malnutrition, infection, drug or alcohol addictions within the past year; schizophrenia, schizoaffective disorder, or primary affective disorder; severe heart, brain, liver, kidney, lung, and hematopoietic system diseases; severe auditory, visual, or motor deficits impairing cognitive testing; and other serious primary diseases.
Table 1. Demographic and clinical parameters of the two groups.
|
|
AD (n=30) |
NC (n=30) |
t/c2 |
P value |
|
Age (years) |
65.17±11.34 |
65.63±7.08 |
0.246 |
0.620 |
|
Female (%) |
14(46.67%) |
13(43.33%) |
0.067 |
0.795 |
|
Education (years) |
9.07±4.08 |
8.47±3.15 |
0.346 |
0.557 |
|
Diabetes (%) |
9(30.0%) |
8(26.67%) |
8.208 |
0.774 |
|
Hypertension (%) |
11(36.67%) |
10(33.33%) |
7.326 |
0.786 |
|
MMSE |
18.17±6.10 |
26.43±1.63 |
35.101 |
<0.001 |
|
MoCA |
13.93±5.27 |
23.70±2.73 |
38.823 |
<0.001 |
|
CDT |
19.96±4.56 |
24.78±2.61 |
116.43 |
<0.001 |
|
CDR |
1.15±0.51 |
0.00±0.00 |
- |
- |
|
CSF Aβ42 (pg/ml) |
452.99±101.25 |
1062.29±101.25 |
-5.495 |
<0.001 |
|
CSF Aβ40 (pg/ml) |
9902.66±2431.47 |
12280.38±1904.07 |
-2.513 |
0.021 |
|
CSF Aβ42/Aβ40 |
0.047±0.011 |
0.086±0.019 |
-6.029 |
<0.001 |
|
CSF t-tau (pg/ml) |
413.27±166.92 |
81.27±82.79 |
5.714 |
<0.001 |
|
CSF p-tau (pg/ml) |
195.1±233.98 |
42.86±16.13 |
2.045 |
0.054 |
|
APOE genotype group |
|
|
3.314 |
0.191 |
|
E 2 |
7(23.3%) |
13(43.3%) |
|
|
|
E 3 |
12(40.0%) |
11(36.7%) |
|
|
|
E 4 |
11(36.7%) |
6(20.0%) |
|
|
Notes: Data are given as means±SD or percentages. Twelve AD patients and 10 controls underwent lumbar puncture, and the typical CSF Aβ42, Aβ40, t-tau and p-tau were tested. There is no e2e4 genotype in this study. Abbreviations: AD, Alzheimer’s disease; NC, normal cognition healthy controls. MMSE, Mini-Mental State Examination; MoCA, Montreal Cognitive Assessment; CDT, clock drawing test; CDR, Clinical Dementia Rating; CSF, Cerebrospinal fluid; APOE, Apolipoprotein E. SD, standard deviation. APOE genotype group defined as E2=ε2ε2 or ε2ε3; E3=ε3ε3; and E4=ε3ε4 or ε4ε4.
2). It seems necessary to describe the presence and types of comorbidities in 30 AD and 30 NC subjects. It was described that most of the biomarkers discovered were related to inflammation, but it is necessary to mention comorbidities to confirm if they are only related to AD pathology.
Answer: Thanks for the reviewer’s suggestion. We added some demographic and clinical parameters of AD and NC groups in the table 1. And add the exclusion criteria in the “Materials and Methods” section.
3).There is no problem with statistical analysis, but mention whether the version of SPSS is too low, and more than that, describe the source of the statistical program.
Answer: Thanks for the carefulness of the reviewer. We used SPSS version 23.0 (IBM Corp., Armonk, NY) to recalculate the data. So, we rewrote the“2.5 Statistical analysis”section as “Statistical analyses were performed using SPSS version 23.0 (IBM Corp., Armonk, NY). The normality of distribution of continuous variables such as age, MMSE score and protein concentration, was tested by one-sample Kolmogorov-Smirnov test. Means of continuous normally distributed variables of the two groups were compared by independent samples Student’s t test. Mann-Whitney U test were used to compare means of variables not normally distributed. Group differences in categorical data, such as sex, and apolipoprotein genotype subgroup, were analyzed using the Chi-square test. Correlative analysis was performed using a linear regression model. To assess the potential prediction efficacy for selected DEPs and in combinations, the receiver operating characteristic (ROC) curve and the area under the ROC curve (AUC) analysis were performed. P < 0.05 was considered as statistically significant.”
4) This study is referred to as "The majority of DEPs in the CSF and serum of AD subjects which were mainly associated with inflammation, ATPase activity, oxidative stress and mitochondrial dysfunction, were significantly correlated with the results of the cognitive assessment.". But it seems that more discussion is needed on the relationship between cognitive decline and the corresponding mechanism in the biomarker found.
Answer: Thanks for the reviewer’s suggestion. It is really true as Reviewer suggested that the inflammation, ATPase activity, oxidative stress and mitochondrial dysfunction, were significantly correlated with cognitive decline. In the discussion section, we discussed the functions of several important differential proteins and their potential relationship with AD, respectively.
“PRDX3 is the only one exclusively localized to mitochondria, which are the main source of reactive oxygen species. Excessive levels of reactive oxygen species are harmful to cells, inducing mitochondrial dysfunction, DNA damage, lipid and protein oxidation and ultimately apoptosis. Neuronal damage induced by oxidative stress and mitochondrial dysfunction is associated with numerous neurodegenerative disorders, such as AD, Parkinson’s disease and cerebellar ataxia[11]. Compared with APP transgenic mice, APP/PRDX3 transgenic mice show less cognitive decline and reduced amyloid beta levels in the brain[12]. PRDX3 knockdown in an AD cell model (N2a-APPswe cells) induced dysregulation of more than one hundred proteins, which were enriched for protein localization to the plasma membrane, the lipid catabolic process, and intermediate filament cytoskeleton organization[13].”.
“SRGN plays an important regulatory role in the inflammatory response of the central nervous system. Heparan sulfate proteoglycans promote amyloid fibril formation and tau fibrillization in AD and provide resistance against proteolytic breakdown[15,16]. Recently, Lorente-Gea et al.[17] investigated the expression of heparan sulfate proteoglycans in various human AD brain areas at different Braak stages. SRGN and SDC4 were over-expressed in most of the areas, and immunohistochemistry revealed the presence of SRGN in all three types of AD lesion: neuritic plaques, cerebral amyloid angiopathy, and neurofibrillary pre-tangles and tangles. SRGN is the only intracellular heparan sulfate proteoglycan; therefore, the authors speculated that it may play a central role in AD stabilization and progression by means of the 3-O-sulfated domains in heparan sulfate polysaccharide chains[17]. Furthermore, significant numbers of the variously expressed proteins are involved in N-glycan biosynthesis according to the functional interpretation provided by KEGG analysis. It is currently considered that N-glycans contribute significantly to the development of AD.”.
“In the central nervous system, Na+-K+-ATPase establishes transmembrane ion gradients and maintains excitability of neurons and glial cells, thereby playing an important role in the process of learning and memory. Dysregulation and deficiency of neuronal Na+-K+-ATPase have been implicated in the pathogenesis of cognitive disorders, including vascular dementia, and AD. Interaction of Aβ oligomers with Na+-K+-ATPase contributes significantly to the development of AD[26]. Na+-K+-ATPase expression is significantly decreased in patients with AD and in a transgenic mouse model of AD[27,28].”
We would like to express our great appreciation to you for comments on our paper. Looking forward to hearing from you.
Thank you and best regards.
Yours sincerely,
Guoping Peng, MD, PhD

Round 2
Reviewer 1 Report
Dear Prof. Guoping Peng,
Thank you for incorporating my suggestions in the manuscript. I recommend acceptance.
Best regards
Reviewer 2 Report
Congratulations.